# *Auricularia auricula* Peptides Nutritional Supplementation Delays H_2_O_2_-Induced Senescence of HepG2 Cells by Modulation of MAPK/NF-κB Signaling Pathways

**DOI:** 10.3390/nu15173731

**Published:** 2023-08-25

**Authors:** Qianwen Han, Haiyan Li, Fen Zhao, Ji’an Gao, Xinqi Liu, Biao Ma

**Affiliations:** 1Key Laboratory of Geriatric Nutrition and Health, Beijing Engineering and Technology Research Center of Food Additives, Beijing Technology and Business University, Beijing 100048, China; hanqianwenya@163.com (Q.H.); l15069433928@163.com (H.L.); gja200828@163.com (J.G.); liuxinqi@btbu.edu.cn (X.L.); 2National Soybean Processing Industry Technology Innovation Center, Beijing Technology and Business University, Beijing 100048, China; 3Beijing Science Sun Pharmaceutical Co., Ltd., Beijing 100176, China; mabiao@ssyy.com.cn

**Keywords:** *Auricularia auricula* peptides, cell senescence, oxidative stress, inflammation, SASP, RNA-seq

## Abstract

*Auricularia auricula* is a traditional medicinal and edible mushroom with anti-aging effects. Many studies focused on polysaccharides and melanin. However, the anti-aging effects and mechanism of the nutritional supplementation of *Auricularia auricula* peptides (AAPs) were not elucidated. In this study, AAPs were prepared by enzymolysis of flavor protease and the protective effects on H_2_O_2_-induced senescence of HepG2 cells were explored for the first time. The potential mechanism was also investigated. AAPs were mostly composed of low molecular weights with less than 1000 Da accounting for about 79.17%, and contained comprehensive amino acids nutritionally, including seven essential amino acids, aromatic, acidic, and basic amino acids. AAPs nutritional supplementation could significantly decrease the levels of intracellular reactive oxygen species (ROS) and malondialdehyde (MDA), and increase the activities of antioxidant enzymes (SOD, CAT, and GSH-Px). In addition, the senescence-associated-β-galactosidase (SA-β-gal) activity was restrained, and the expression levels of senescence-associated secretory phenotype (SASP) (IL-6, IL-8, IL-1β, and CXCL2) were also decreased. Ribonucleic acid sequencing (RNA-Seq) was carried out to screen the differentially expressed genes (DEGs) between different groups. GO and KEGG enrichment analysis showed that the mechanism was related to the MAPK/NF-κB signaling pathways. Quantitative real-time PCR (qRT-PCR) analysis and Western blot were carried out to verify the key genes and proteins in the pathways, respectively. AAPs nutritional supplementation resulted a significant down-regulation in key the genes c-fos and c-jun and up-regulation in DUSP1 of the MAPK signaling pathway, and down-regulation in the key genes CXCL2 and IL-8 of the NF-κB signaling pathway. The results of Western blot demonstrate that AAPs nutritional supplementation could inhibit MAPK/NF-κB pathways by reducing the expression levels of IKK, IκB, P65, and phosphorylation of ERK, thus decreasing the inflammatory reaction and delaying cell senescence. It is the first time that AAPs nutritional supplementation was proved to have protective effects on H_2_O_2_-induced oxidative damage in HepG2 cells. These results implicate that dietary AAPs could be used as nutrients to reduce the development or severity of aging.

## 1. Introduction

Bioactive peptides are usually a mixture of protein hydrolysate and low molecular weight peptides [1]. So far, bioactive peptides were isolated from various animals and plants. Bioactive peptides from edible fungi are more easily absorbed and available than those from animals. The bioactive peptides of edible fungi can be completely absorbed by the intestines without increasing the functional burden of the gastrointestinal tract [2]. Many edible fungi, such as *Ganoderma lucidum*, *Agaricus bisporus*, and *Auricularia auricula*, have antioxidant and anti-aging activities [3,4,5]. *Auricularia auricula* is a traditional medicinal and edible mushroom. *Auricularia auricula* is the second-highest yielding edible fungus in China and is deeply loved by consumers. *Auricularia auricula* is rich in polysaccharide, protein, crude fiber, vitamin B, and trace elements such as potassium, iron, and calcium. *Auricularia auricula* has antioxidant, anti-aging, anti-inflammatory, anticoagulation, hypolipidemic, antibacterial activities, liver protection, and antitumor activities [6]. Studies showed that increasing the activities of enzyme antioxidants (POD, CAT, and SOD) and non-enzymatic antioxidants (GSH and AsA) in *Auricularia auricula* could effectively eliminate ROS without damaging the cell membrane [7]. *Auricularia auricula* peptides (AAPs) have an ameliorative effect on reducing MDA levels and increasing SOD levels in HepG2 cells [8]. However, the anti-aging activity and the mechanism of *Auricularia auricula* peptide nutritional supplementation were not reported.

Aging is a complex process that affects all organs [9,10]. The age-induced decline in organ functions involves several cell-autonomous events termed the “hallmarks of aging” [11]. Twelve hallmarks of aging were proposed, and oxidative stress and inflammation were two important ones [12]. Oxidative stress is extremely related to inflammation. The process called inflamm-aging is a vital foundation for cell senescence, the aging process, and frailty in humans, particularly in the development of age-related disorders [13]. Over the past two decades, efforts to identify anti-aging interventions that reduce morbidity and increase life span intensified [11]. However, considering the side effect of the drugs, nutrient supplements were considered as one efficient way.

ROS are free radicals and peroxides generated by cells in the process of normal metabolism and are essential for processes, including cytokine transcription, ion transport, and immune regulation [14]. ROS with lower levels could work as a signaling factor and affect the expression of relevant genes, which could be regulated through the antioxidant system [15]. However, excessive production of ROS can have damaging effects by oxidizing biologically essential molecules such as lipids, carbohydrates, proteins, and DNA, thereby causing oxidative damage to cell membranes and tissues, which may ultimately lead to aging-related diseases [16]. The end product of lipid peroxidation is MDA, which can cause molecular cross-polymerization on the biological scale. The antioxidant enzyme system in organisms mainly includes CAT, SOD, and GSH-Px, which can maintain the balance between oxidants and antioxidants in the organism [17]. SA-β-gal activity is the first reported marker related to lysosomal activity and the most commonly used to evaluate cell senescence [18]. β-galactosidase is optimal at a low pH of 4.0–4.5 and is significantly magnified as a result of the increasing of lysosomal content in senescent cells. As a consequence, the detection of β-galactosidase activity at pH 6.0 (above optimal pH) is essential to the specific recognition of senescent cells [19]. Senescent cells secrete redundant factors, including chemokines, pro-inflammatory cytokines, matrix metalloproteinases, growth regulators, and angiogenic factors, collectively known as the SASP [20,21]. Senescent cells promote the production of SASP factors and eventually induce inflammation and senescence of neighboring cells [22]. However, there was a certain degree of overlap in various SASP factors, and specific proteins were almost always found, such as IL-8, IL-6, IL-1, and so on [23]. The SASP factors are regulated by many elements in senescent cells, including the activation of signaling pathways and transcription factors such as NF-κB [24,25]. SASP is released more readily when cells are under oxidative stress. The NF-κB signaling pathway, one of the most crucial signaling pathways in the inflammatory response, is activated by the interaction of SASP factors and cell surface receptors. Nuclear transcription factor NF-κB plays a significant role in upstream inflammation. The downstream IKK activates IκB phosphorylation, which promotes the translocation of NF-κB into the nucleus and thus the secretion of proinflammatory cytokines (IL-6 and IL-1β) [26]. In turn, IL-1 activates NF-κB and initiates IL-1 transcription, forming a loop and aggravating inflammatory injury. Therefore, the study of oxidative stress and inflammatory reaction is very important to explore cell senescence. Extracellular mediums regulate NF-κB by activating various signaling cascades associated with inflammation. Among them, mitogen-activated protein kinase (MAPK) is a vital signaling transduction molecule for the production of inflammatory factors. The phosphorylation of the MAPK signaling pathway mainly includes JNK1/2, p38 MAPK, and ERK1/2 subunits, which participate in the regulation of upstream NF-κB, leading to the expression and activity of inflammatory factors [27,28].

As one of the three commonly used cell lines for anti-aging studies, HepG2 is a well-differentiated transformed cell line with good characteristics, which is reliable, easy to cultivate, and widely used in biochemical and nutritional research [29,30]. HepG2 cells have a stronger steady-state antioxidant defense activity than hepatocytes and other non-transgenic cells, and it is easier to detect changes in response to different conditions. The oxidative damage model of HepG2 cells is extensively applied to the cell-based bioassay of food antioxidant activity. H_2_O_2_-induced HepG2 could establish a reliable cell senescence model caused by oxidative damage [31]. *Auricularia auricula* was known to be beneficial for the liver in traditional Chinese medicine. However, so far, the active components of liver protection were not known, and HepG2 cells were chosen to see if *Auricularia auricula* peptides had good protective effects on the liver simultaneously. Thus, above all, HepG2 cells were chosen as the model cells.

Based on the above information, we hypothesized that AAPs nutritional supplementation might delay cell senescence by reducing oxidative stress and inflammation. It is the first time that the protective effects of AAPs nutritional supplementation on H_2_O_2_-induced HepG2 cell senescence and molecular mechanisms were investigated.

## 2. Materials and Methods

### 2.1. Materials and Chemicals

*Auricularia auricula* was purchased from Heilongjiang, China. The human hepatoma (HepG2) cells were obtained from the Institute of Basic Medical Sciences, Chinese Academy of Medical Sciences (Beijing, China). Minimum essential medium (MEM), non-essential amino acids (NEAA), penicillin-streptomycin (PS), and trypsin-EDTA (TE) were obtained from Gibco (Grand Island, NE, USA). Fetal bovine serum (FBS) was obtained from Excell Biotechnology Co. (Taicang, China). H_2_O_2_ was purchased from Sigma Chemical Co. (St. Louis, MO, USA).

### 2.2. Preparation of AAPs

Dried *Auricularia auricula* was crushed into powder and passed through the 100-mesh screen. As demonstrated in Figure 1, *Auricularia auricula* protein was isolated by alkaline solubilization and acid deposition methods. Alkaline solubilization was carried out at 40 °C for 2 h at pH 9 and the solid–liquid ratio was 1:80. After centrifuging at 3300× *g* for 15 min, the supernatant was adjusted to pH 3.5. The precipitated protein was collected and dissolved in water to confirm that the mass fraction of a solution was 8%. Flavor protease was used for enzymolysis at 45 °C for 2 h to obtain AAPs. The protease was inactivated by heating at 95 °C for 15 min. Ethanol was added while stirring until the final concentration was 60%. Ultimately, the degree of hydrolysis reached 60%. The supernatant was collected after centrifugation and filtered through a 0.45 μm microporous membrane. The AAPs were collected after freeze-drying.

The protein content, amino acid composition, and molecular weight were determined by using the previous method with some modifications [1]. The protein content of AAPs was measured using a conversion factor of 6.25 through the Kjeldahl method (Kjeltec 8000, FOSS analysis A/S, Hilleroed, Denmark). The amino acid composition of AAPs was determined in three replicate samples by an amino acid analyzer (Biochrom 30+ amino acid analyzer, Biochrom Co., Ltd., Cambridge, UK) after hydrolysis with 6 N HCl (110 °C) on a Na cation exchange column (8 μm, 4.6 × 200 mm) for 24 h. Gel filtration chromatography (GFC) with a Shimadzu LC20A instrument was used to determine the molecular weight distribution of AAPs. The column was the TSK-GEL G2000SWXL column (5 μm, 7.8 × 300 mm).

### 2.3. Cell Culture and Treatments

The cells were cultured in MEM supplemented with 10% FBS, 1% NEAA, and 1% PS in a humidified incubator containing 5% CO_2_ at 37 °C.

The H_2_O_2_-induced HepG2 senescent model was established in a manner previously described [32]. HepG2 was incubated for 24 h in the FBS-free MEM supplemented with different concentrations of H_2_O_2_ (from 50 to 1000 µM). The cell viability and ROS levels were estimated to screen the effective intervention dose of H_2_O_2_.

### 2.4. Cell Viability Assay

The effects of H_2_O_2_ and AAPs nutritional supplementation on HepG2 cell viability were measured by the CCK-8 assay. HepG2 cells were treated with different concentrations of AAPs (0.00–4.00 mg/mL) or H_2_O_2_ (0–1000 μM) and preincubated for 24 h in the incubator. The 10 µL CCK-8 solution was added to each well and cultured for 1 h in 96-well plates. Ultimately, the OD values were recorded with a microplate reader (Infinite 200 Pro Nanoquant, Tecan, Männedorf, Switzerland) at 450 nm, and the cell viability was calculated based on the instructions.

### 2.5. Determination of ROS Induced by H_2_O_2_ and the Protective Effect of AAPs Nutritional Supplementation

The level of ROS caused by H_2_O_2_ were measured using the 2′,7′-dichlorofluorescein diacetate (DCFH-DA) assay [33]. The protective effect of AAPs nutritional supplementation on HepG2 cells senescence was determined by adding H_2_O_2_ after 24 h of AAPs nutritional supplementation. Briefly, 10 μM DCFH-DA in MEM (FBS-free) was added to a 6-well plate for culturing HepG2 cells and incubated at 37 °C for 30 min, and then washed with MEM three times to remove the residuary DCFH-DA. After labeling with DCFH-DA, the fluorescence intensity of the cells was measured by an enzyme-labeled instrument at 488 nm excitation wavelength and 525 nm emission wavelength.

### 2.6. Determination Activities MDA and Antioxidant Enzymes

The levels of MDA, SOD, CAT, and GSH-Px in cells were measured according to the instructions of the manufacturer of the commercial kits. The protein concentration of HepG2 cells was determined by the BCA kit.

### 2.7. Senescence-Associated β-Galactosidase (SA-β-Gal) Staining

The senescent cells can be specifically identified by the determination of SA-β-gal [34]. The SA-β-gal staining experiment was carried out according to the manufacturer’s instructions using the β-Galactosidase staining kit (Solarbio, Beijing, China). The staining images were taken on a microscope (IX73, OLYMPUS, Tokyo, Japan) and the percentage of SA-β-gal-positive cells was calculated manually.

### 2.8. Senescence-Associated Secretory Phenotype (SASP)

After AAPs and H_2_O_2_ treatment, cell-free supernatants were collected. The concentrations of IL-1β, IL-6, IL-8, and CXCL2 in the cell culture liquid were measured using commercial ELISA quantification kits based on the manufacturer’s specifications. The OD values of the solutions were determined at 450 nm with the help of an automated microplate reader.

### 2.9. Library Construction and Sequencing

The Trizol method was used to extract total RNA [35]. The NanoDrop 2000 spectrophotometer was used to determine the purity and concentration of total RNA (Thermo Fisher Scientific, Waltham, MA, USA). Illumina’s TruSeq^TM^RNA sample preparation kit (San Diego, CA, USA) was used to build an RNA-seq transcription library from 2 μg of total RNA. The Illumina Novaseq 6000 (2 × 150 bp read length) was used to sequence the paired-end RNA-seq sequencing library. The raw RNA-seq data were deposited in the Sequence Read Archive (SRA) database at the National Center of Biotechnology Information (NCBI) with the accession numbers SRR23147761–SRR23147769.

### 2.10. Quantitative Real-Time PCR (qRT-PCR)

The Transzol up plus RNA kit was used to extract total RNA from the samples before being transcribed into the first-strand cDNA using a PerfectStart uni RT & qPCR kit. The genes with significantly differential multiples were selected for a qRT-PCR experiment to confirm the correctness of the transcriptome data. The expression levels of selected genes were determined by the 2^−ΔΔCt^ method. The forward and reward primer sequences of genes were shown in Table 1. The ACTB gene was used as an internal control. The reaction was performed on a real-time PCR detection system (Bio-Rad, Hercules, CA, USA) using PerfectStart uni RT & qPCR kit.

### 2.11. Western Blot

HepG2 cells were collected and lysed on ice with the lysate (RIPA). The protein concentrations of all samples were determined and unified by the BCA method. The loading amount of each protein sample was 10 μL and was transferred to the PVDF membrane after 10% SDS-PAGE electrophoresis. After sealing with 5% skimmed milk, the first antibody was added and incubated overnight at 4 °C, and then the first antibody was washed away. The second antibody was incubated at room temperature for 1 h. Finally, the ECL luminescent substrate was used for Western blotting. Use Image J software to analyze the band intensities.

### 2.12. Statistical Analysis

All data were analyzed by a one-way analysis of variance (ANOVA) test using SPSS (version 24) software. All experiments were carried out independently and repeated three times, and *p* < 0.05 was considered as a significant difference.

## 3. Results

### 3.1. Determination of the Properties of AAPs

The protein content, amino acid composition, and molecular weight of AAPs were determined. Amino acid composition is closely related to antioxidant activity [36]. Aromatic amino acids can effectively scavenge free radicals due to their aromatic structures and the presence of phenolic groups [37]. Acidic and basic amino acid residues can effectively chelate metal ions and basic amino acids are hydrogen donors with strong free radical scavenging activity [38,39]. The protein content of AAPs was 62.62%. As shown in Table 2, AAPs contained comprehensive amino acids, including seven essential amino acids, aromatic, acidic, and basic amino acids. Aromatic amino acids including tyrosine, phenylalanine, and tryptophan, acidic amino acids including aspartate and glutamate, and basic amino acids including lysine, arginine, and histidine accounted for 7.7%, 28.46%, and 18.00% of the total amino acids, respectively, which indicated that AAPs had strong antioxidant capacity. Bioactive peptides are usually a mixture of protein hydrolysate and low molecular weight peptides. Tao et al. reported that *Auricularia auricula* peptides (molecular weight less than 3 kDa) increased the activity of intracellular SOD, and decreased the accumulation of MDA in HepG2 cells, which alleviated the oxidative stress reaction to some extent [40]. As described in Figure 2, the molecular weights distributed for less than 1000 Da were about 79.17% and for 1000–3000 Da were about 11.93%, which indicated good antioxidant activity. AAPs are mixed peptides, which may lead to low peak resolution. The determination of molecular weight is to obtain the content of peptide segments with different molecular weight ranges according to the peak time and peak area. Therefore, the resolution of the peak has little effect on the molecular weight value.

### 3.2. Evaluation of Cell Viability and ROS Production by H_2_O_2_

Cell viability is an important indicator of cell survival under toxic substances or pressure. The effects of different concentrations of H_2_O_2_ (50–1000 μM) and AAPs (0.25–4 mg/mL) on HepG2 cell viability were determined by the CCK-8 assay. As described in Figure 3A, the viability of HepG2 cells decreased gradually as the concentration of H_2_O_2_ increased. When the concentration of H_2_O_2_ was greater than 100 μM, the cell viability decreased significantly (*p* < 0.05). After 24 h of treatment with 150 μM H_2_O_2_, the cell viability reached 63.44%. The ROS level of different H_2_O_2_ concentrations on HepG2 cells was also determined. The intracellular oxidative stress in HepG2 cells was positively related to the H_2_O_2_ concentration when it was higher than 150 μM (*p* < 0.05, Figure 3B). Thus, 150 μM was selected as the ultima concentration of oxidative stress caused by H_2_O_2_.

As shown in Figure 3C, the viability of HepG2 cells reduced as the AAPs concentration increased. When the AAP concentration was 0.25 mg/mL, the cell viability was significantly increased by 3.35% compared with the control (*p* < 0.05). The cell viability decreased by 1.20%, 1.58%, and 1.92%, respectively, when the concentration was 0.5, 1, and 2 mg/mL, which has no significant difference compared with the control. Moreover, the HepG2 cell viability reduced significantly (*p* < 0.05) when the concentration was higher than 3 mg/mL, indicating that the HepG2 cells were damaged severely. Therefore, the 0.5, 1, and 2 mg/mL were selected as low (AAPs-L), medium (AAPs-M), and high (AAPs-H) concentrations of AAPs to further research the protective effects of different concentrations of AAPs on H_2_O_2_-induced senescence of HepG2 cells.

### 3.3. Preventive Effects of AAPs Nutritional Supplementation on Cell Viability and Reduction in Intracellular ROS

As shown in Figure 4A, compared with the model group, the cell viability of all the AAPs nutritional supplementation groups (0.5, 1, and 2 mg/mL) increased significantly, which was 7.53%, 20.14%, and 48.34%, respectively. The effect of AAPs-H group was similar with the control group, and had no significant difference compared with the control group. The results indicate that AAPs nutritional supplementation had remarkable protective effects on H_2_O_2_-induced oxidative damage. As shown in Figure 4B,C, H_2_O_2_ stimulation led to a dramatic increase in ROS fluorescence intensity. However, compared with the model group, all the AAPs nutritional supplementation groups could significantly reduce intracellular ROS fluorescence intensity by 48.59%, 53.77%, and 69.06%, respectively. The results demonstrate that AAPs nutritional supplementation could protect HepG2 cells from H_2_O_2_-induced oxidative damage and eliminate the accumulation of intracellular ROS.

### 3.4. Effect of AAPs on Oxidative Damage in HepG2 Cells

In our study, the levels of CAT, SOD, GSH-Px, and MDA in the HepG2 cells were determined to estimate the alleviative effect of AAPs nutritional supplementation on H_2_O_2_-induced oxidative damage. As shown in Figure 5, MDA in the model group was 109.59% higher than the control group, indicating stronger oxidative damage in the HepG2 cells after H_2_O_2_ stimulation. However, all the AAPs nutritional supplementation groups significantly decreased (*p* < 0.05) by 14.29%, 24.51%, and 47.06%, respectively. Additionally, the AAPs-H nutritional supplementation group even had no significant difference with the control group, which showed obvious protective effect. Comparing with the control group, the model group significantly decreased (*p* < 0.05) the CAT, SOD, and GSH-Px contents in the HepG2 cells by 46.13%, 42.58% (*p* < 0.01), and 41.20%, respectively, which also indicated serious strong oxidative damage after H_2_O_2_ treated. Compared with the model group, all the AAPs nutritional supplementation groups could significantly improve the CAT, SOD, and GSH-Px levels. Surprisingly, the CAT and SOD contents in AAPs-H nutritional supplementation group, and GSH-Px content in AAPs-M and AAPs-H nutritional supplementation groups all had no significant differences with the control group, which showed strong antioxidant protective activities.

### 3.5. Senescence-Associated β-Galactosidase (SA-β-Gal) Staining

SA-β-gal activity is an essential marker of senescence. It can be detected in most aging environments, but not in proliferating or resting cells. The number of SA-β-gal-positive cells can indicate the degree of cell senescence. As shown in Figure 6A, the cells were significantly larger and flatter in the model group, which indicated that the cell senescence model was successfully established. The positive staining rate of SA-β-gal in the model group was significantly increased by 139.34% (*p* < 0.05) after H_2_O_2_ treatment compared with the control group (Figure 6B). Compared with the model group, the SA-β-gal-positive staining rate of AAPs-L, AAPs-M, and AAPs-H nutritional supplementation groups significantly decreased (*p* < 0.05) by 34.58%, 40.60%, and 54.98% with the increasement of AAPs concentration, which indicated that the nutritional supplementation with AAPs could improve the condition of senescent cells. In particular, the AAPs-H nutritional supplementation group had no significant difference with the control group, which indicated that AAPs nutritional supplementation could inhibit cell senescence induced by oxidative stress through reducing the number of senescent cells and decreasing the accumulation of senescent cells.

### 3.6. Senescence-Associated Secretory Phenotype (SASP)

As shown in Figure 6C–F, the levels of typical SASP including IL-1β, IL-8, IL-6, and chemokine CXCL2 were measured. The secretion levels of SASP in the model group significantly increased (*p* < 0.05) by 95.53%, 58.13%, 34.05%, and 23.73%, respectively, compared with the control group. However, after nutritional supplementation with different concentrations of AAPs, the levels of SASP all dramatically decreased with the increase in AAPs concentration, which indicated that nutritional supplementation with AAPs could dramatically reduce the levels of SASP. Additionally, the AAPs-H nutritional supplementation group showed the best inhibitory effect on IL-1β, IL-8, IL-6, and CXCL2, which significantly reduced (*p* < 0.05) by 33.15%, 35.27%, 25.35%, and 16.60%, respectively. Notably, the expression levels of IL-8 in the AAPs-M and AAPs-H nutritional supplementation groups, IL-1β in the AAPs-H nutritional supplementation group, and CXCL2 in the AAPs-L, AAPs-M, and in the AAPs-H nutritional supplementation groups all had no significant differences with the control group.

### 3.7. Transcriptome Sequencing and Gene Expression Analysis

Based on the above results, the AAPs-H group showed the most effective protection activity. Therefore, the AAPs-H group was selected to perform RNA-seq with the control and model groups. The original reads obtained from the cDNA libraries constructed from the control, model, and AAPs-H nutritional supplementation groups were 59,125,972, 61,810,745, 54,768,897, respectively. The results of both Q20 and Q30 are above 90% (Table 3), indicating good quality and high accuracy of the transcriptome sequencing data. The overall gene changes were inspected by correlation analysis and differential expression of DEGs is shown in Figure 7. As demonstrated in Figure 7A, the correlation coefficients between any two samples in the group were all greater than 0.9, indicating that the three samples were biologically repeatable and the experimental design was rational. A total of 320 DEGs (fold-change > 2 and *p* < 0.05) were identified in the model/control group with 155 up-regulated and 165 down-regulated genes (Figure 7B,C). A total of 472 DEGs were identified in the AAPs-H nutritional supplementation/control group with 216 up-regulated and 256 down-regulated genes (Figure 7B,D). A total of 228 DEGs were identified in the AAPs nutritional supplementation/control group including 125 up-regulated and 103 down-regulated genes (Figure 7B,E).

### 3.8. Gene Ontology (GO) Enrichment Analysis

To investigate the mechanism of nutritional supplementation with AAPs on H_2_O_2_-induced senescence in HepG2 cells, GO and KEGG enrichment analyses were carried out on DEGs (Figure 8). In the model/control group, 329, 103, and 34 GO terms were significantly enriched in the biological process (BP), molecular function (MF), and cellular component (CC), respectively (Figure 8A). Additionally, the regulation of cell migration, negative regulation of phosphorus metabolic process, cellular response to stimulus, regulation of the natural killer cell apoptotic process, cellular response to tumor necrosis factor, serotonin secretion involved in inflammatory response, regulation of hydrogen peroxide-induced neuron death, chemokine-mediated signaling pathway, and the cell cycle process were the most significant items in the BP section. In the MF section, the protein activity was mainly associated with protein binding and chemokine activity. In the CC section, DEGs are mainly associated with transcription factor AP-1 complex and IκB kinase complex. The GO terms in the model/control group indicated that H_2_O_2_ oxidative stress stimulation might activate the inflammatory pathway. In the AAPs-H nutritional supplementation/control group (Figure 8B), 695, 141, and 81 GO items were enriched in BP, MF, and CC, respectively. The regulation of ERK1 and ERK2 cascade, the regulation of the multicellular organismal process and developmental process, and the regulation of cell population proliferation were the main items enriched in the BP section. Protein binding, signaling receptor binding, and the structural constituent of the cytoskeleton were the main items enriched in the MF section. The CC section is mainly related to extracellular space, cell periphery, and membrane-bounded organelles. These GO items were different from the model/control group and related to cell proliferation and the MAPK signaling pathway. Moreover, DEGs enriched in protein binding and cAMP response element binding in the MF section were the same as that in model/control. In the AAPs-H nutritional supplementation/model group (Figure 8C), 129, 58, and 15 GO items were enriched in BP, MF, and CC, respectively. Items such as positive regulation of the cellular response to hepatocyte growth factor stimulus, regulation of the hepatocyte growth factor biosynthetic process, Ras protein signal transduction, inflammatory response, negative regulation of secretion by cell, and the *N*-acylphosphatidylethanolamine metabolic process were the significantly enriched GO categories in the BP section. The MF part is mainly enriched in host cell surface receptor binding, DNA binding, and bending. The CC part is mainly related to substances such as troponin complex, central element, basement membrane, BLOC-1 complex, and the interstitial matrix. The GO items in the AAPs nutritional supplementation/model group indicated that AAPs had a positive regulation of the cellular response to hepatocyte growth factor stimulus and had an effect on inflammation-related protein signaling transduction.

### 3.9. Kyoto Encyclopedia of Genes and Genomes (KEGG) Enrichment Analysis

The KEGG pathway enrichment analysis of the DEGs in different groups is demonstrated in Figure 9. The pathways in different groups were investigated according to the crucial pathways (*p* < 0.05) to better understand the mechanism of AAPs-H nutritional supplementation on delaying cell senescence. DEGs were significantly enriched in the NF-κB signaling pathway, IL-17 signaling pathway, TNF signaling pathway, Toll-like receptor signaling pathway, NOD-like receptor signaling pathway, and MAPK signaling pathway, which were all associated with inflammation. The NOD-like receptor signaling pathway, TNF signaling pathway, Toll-like receptor signaling pathway, and IL-17 signaling pathway were significantly enriched in both model/control and AAPs-H nutritional supplementation/control groups. These three signaling pathways were considered as the link between inflammatory disorders and the innate immune system. The IL-17 family in the IL-17 signaling pathway activates anti-cytokines and chemokines in the NF-κB and MAPK signaling pathways, which are early initiators of T cell-induced inflammatory responses and can amplify inflammatory responses by promoting the release of pro-inflammatory cytokines [41]. Therefore, their effects on H_2_O_2_-induced senescence were mostly auxiliary. According to the conclusions of KEGG pathway enrichment analysis, NF-κB and MAPK signaling pathways were two key pathways for AAPs to prevent HepG2 cell senescence. The NF-κB signaling pathway was chiefly enriched in the model/control group, the MAPK signaling pathway was chiefly enriched in the AAPs nutritional supplementation/control group, and cell senescence was the main enriched signaling pathway in the AAPs nutritional supplementation/model group. Thus, KEGG pathway enrichment analysis demonstrated that the effect of AAPs on cell senescence was closely related to the MAPK/NF-κB inflammatory signaling pathways.

### 3.10. Quantitative Real-Time PCR (qRT-PCR) Validation

qRT-PCR was used to validate the RNA-seq data to ensure its reliability. Based on the RNA-seq data from the model/control, AAPs nutritional supplementation/control, and AAPs nutritional supplementation/model groups, twelve DEGs were selected, respectively, for qRT-PCR. Figure 10A–D illustrates the expression levels and correlation analysis of these DEGs. The results demonstrate that the qRT-PCR expression levels were significantly correlated with the RNA-seq data (R = 0.7857) and concurred with the RNA-seq results.

To further determine the regulation of AAPs on cell senescence through the NF-κB/MAPK signaling pathway, the mRNA expression levels of genes in this pathway were measured. Two genes, CXCL2 and IL-8, were related to the NF-κB pathway and three genes (c-fos, c-jun, and DUSP1) were related to the MAPK pathway. As described in Figure 10E, comparing with the control group, four genes (CXCL2, IL-8, c-fos, and c-jun) were significantly up-regulated and one gene DUSP1 was obviously down-regulated in the model group. However, comparing with model group, four genes (CXCL2, IL-8, c-fos, and c-jun) were significantly down-regulated and one gene DUSP1 was obviously up-regulated in the AAPs nutritional supplementation group, which indicated the excellent protective effects of AAPs nutritional supplementation. Remarkably, the expression levels of genes in the AAPs nutritional supplementation group were almost close to the control group with no significant difference.

### 3.11. Western Blot Assay

Western blot was used to determine the expression levels of proteins in the MAPK/NF-κB signaling pathway. As shown in Figure 11A–G, compared with the control group, the NF-κB signaling pathway was highly activated in the model group, and the expression levels of proteins IKK, IκB, and P65 were significantly increased by 79.72%, 36.80%, and 34.09%, respectively. Compared with the model group, the AAPs-L, AAPs-M, and AAPs-H nutritional supplementation groups could significantly inhibit the expressions of IKK by 42.52%, 62.70%, and 68.07% (*p* < 0.05), and significantly inhibit the expression of IκB by 31.14%, 25.82%, and 29.28%, respectively. The AAPs-H nutritional supplementation group significantly inhibited the expression of P65 by 21.32% (*p* < 0.05). Comparing with the control group, all the AAPs nutritional supplementation groups had no significant differences in expressions of IKK, IκB, and P65, except that IKK and ERK in the AAPs-H group were significantly inhibited by 42.61% and 29.68%, respectively. Compared with the control group, the MAPK/ERK signaling pathway was activated in the model group, and the expression levels of ERK and *p*-ERK proteins were significantly increased by 21.51% and 116.30%, respectively. AAPs-M and AAPs-H nutritional supplementation groups could significantly inhibit the expression of ERK protein, and all the AAPs nutritional supplementation groups could significantly inhibit the expression of *p*-ERK protein. Compared with the model group, the AAPs-H nutritional supplementation group showed the best inhibitory effect, and ERK and *p*-ERK were significantly reduced by 42.12% and 65.98%, respectively.

Above all, the results show that AAPs nutritional supplementation could reduce inflammatory response by inhibiting the expressions of IKK, IκB, P65, ERK, and *p*-ERK.

## 4. Discussion

In this study, AAPs contained comprehensive amino acids, including seven essential amino acids, aromatic, acidic, and basic amino acids nutritionally, which indicated that AAPs had strong antioxidant capacity. Bioactive peptides are usually a mixture of protein hydrolysate and low molecular weight peptides. The molecular weights of AAPs distributed for less than 1000 Da were about 79.17%, which also indicated good antioxidant activity.

The injury induced by ROS can expedite cell senescence, and the excessive accumulation of senescent cells promotes the aging of organisms. In this study, the relationship between oxidation, inflammation, and cell senescence was explored, and the mechanism of AAPs nutritional supplementation delaying H_2_O_2_-induced senescence of HepG2 cells was also investigated. Our results show decrease in antioxidant enzyme activities and increasement in lipid peroxidation after H_2_O_2_ stimulation, consequently causing oxidative stress. Cells treated with H_2_O_2_ exhibited a senescent phenotype such as flattened cell morphology, increased cell volume, and raised SA-β-gal activity, which was consistent with previous studies [42]. Amakye et al. found that soybean protein hydrolysate has a more remarkable curative effect than the antioxidant L-glutathione in reversing neurodegeneration related to aging [43]. In our research, AAPs nutritional supplementation could alleviate the H_2_O_2_-induced oxidative damage of HepG2 cells by reducing MDA content and increasing the activities of CAT, SOD, and GSH-Px. Additionally, AAPs nutritional supplementation significantly reduced the activity of SA-β-gal. In addition, the results demonstrate that AAPs nutritional supplementation could inhibit the expression of SASP (IL-1β, IL-6, IL-8, and CXCL2) produced by senescent HepG2 cells. Therefore, the anti-aging effect of bioactive peptides may be better than that of anti-aging agents.

The sequencing results show that the DEGs in the model/control group were significantly enriched in the biological processes of protein binding and response to stimulus. The pathways of the DEGs were chiefly enriched in the inflammation-related signaling pathway. The results prove that H_2_O_2_ stimulation promoted the binding of extracellular inflammatory factors with cell surface receptors. Moreover, the GO enrichment results show that the main GO items in the AAPs/control group were mainly related to biological processes such as regulation of cell population proliferation and developmental process, which was different from the results in the model/control group. AAPs nutritional supplementation could significantly improve the cell growth ability and had the potential to improve the senescent status of HepG2 cells. In the AAPs/model group, GO items such as positive regulation of the cellular response to hepatocyte growth factor stimulus, regulation of the hepatocyte growth factor biosynthetic process, Ras protein signal transduction, inflammatory response, negative regulation of secretion by cell, and the *N*-acylphosphatidylethanolamine metabolic process showed that AAPs nutritional supplementation reduced the inflammatory reaction of senescent HepG2 cells, promoted the growth of HepG2 cells, and the item *N*-acylphosphatidylethanolamine metabolic process was related to neurodegenerative diseases [44].

The qRT-PCR data demonstrate that AAPs nutritional supplementation resulted in a significant down-regulation in the key genes c-fos and c-jun as well as up-regulation in DUSP1 of the MAPK signaling pathway and down-regulation in the key genes CXCL2 and IL-8 of the MAPK signaling pathway. Western blot results show that AAPs nutritional supplementation could decrease inflammatory response by inhibiting the expressions of IKK, IκB, P65, ERK, and *p*-ERK in MAPK/NF-κB signaling pathways. The signaling pathway depends on protein phosphorylation and eventually leads to the activation of specific transcription factors, thus inducing the expression of appropriate target genes. IκB is one of the cytoplasmic inhibitors of NF-κB, and its phosphorylation can regulate NF-κB. IκB is mediated by IκB kinase (IKK), which allows nuclear translocation and transcriptional activation of NF-κB [45]. It is reported that NF-κB can drive the expression of inflammatory genes during aging. CXCL2 and IL-8 were two key down-regulated genes in the NF-κB pathway; they could inhibit the phosphorylation of IKK and IκB and the activation of NF-κB and prevent NF-κB from entering the nucleus to combine with DNA, thereby inhibiting the transcription of proinflammatory factors and reducing the inflammatory reaction [46]. NF-κB is considered the key regulator of SASP, and P65 was proven to inhibit SASP [47,48]. Our study found that AAPs nutritional supplementation significantly inhibited the expression of IKK, IκB, and P65 and reduced the secretion of SASP, reducing the inflammatory response of aging cells. The MAPK/ERK pathway is widely involved in all stages of cell growth and development, including cell proliferation, differentiation, migration, aging, and apoptosis. There is evidence that the ERK signaling plays a key role in cell senescence [49]; c-fos and c-jun were two key genes in the MAPK pathway, and they together composed the heterodimer AP-1. Activation of c-fos and c-jun genes could lead to the activation of ERK1/2 in the MAPK signaling pathway and the production of transcription factor AP-1. DUSP1 could inhibit the activation of the MAPK pathway and deeply reduce IL-1β-induced inflammatory gene expression [50]. AAPs nutritional supplementation significantly inhibited the phosphorylation of ERK, prevented the MAPK signaling pathway, and delayed cell senescence.

In the future, the sequences of peptide segments in AAPs on delaying cell senescence will be studied. Additionally, different cell lines will also be considered to show the effects of AAPs and peptide segments.

## 5. Conclusions

In conclusion, the AAPs prepared in this study were mostly composed of low molecular weight with less than 1000 Da accounting for about 79.17%, and contained comprehensive amino acids, including seven essential amino acids, aromatic, acidic, and basic amino acids, which indicated AAPs had strong antioxidant capacity. AAPs nutritional supplementation could significantly decrease the intracellular ROS and MDA levels and increase antioxidant enzyme activities. In addition, SA-β-gal activity was restrained, and the expression levels of SASP were also decreased. GO and KEGG enrichment analysis showed that the mechanism was related to MAPK/NF-κB signaling pathways, and qRT-PCR analysis verified the expressions of key genes in MAPK and NF-κB signaling pathways. AAPs nutritional supplementation resulted in a significant down-regulation in the key genes c-fos and c-jun and up-regulation in DUSP1 of the MAPK signaling pathway, and down-regulation in the key genes CXCL2 and IL-8 of the NF-κB signaling pathway. Western blot results further demonstrate that AAPs nutritional supplementation could inhibit MAPK/NF-κB signaling pathways by reducing the expression levels and phosphorylation of IKK, IκB, P65, ERK, and *p*-ERK, thereby delaying cell senescence related to inflammation. Our study provided a solid basis for AAPs nutritional supplementation to prevent H_2_O_2_-induced senescence of HepG2 cells for the first time, and AAPs could be used as anti-aging nutrients.

## Figures and Tables

**Figure 1 nutrients-15-03731-f001:**
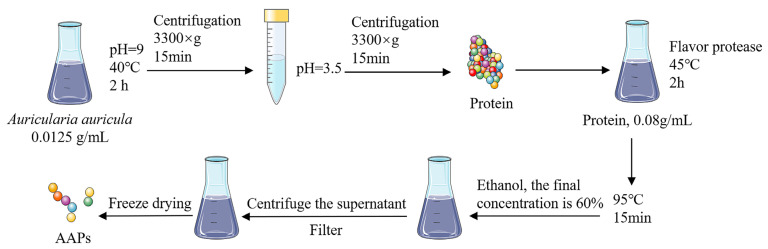
Preparation of AAPs.

**Figure 2 nutrients-15-03731-f002:**
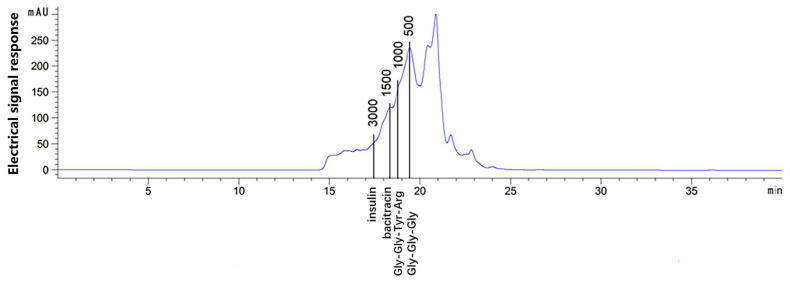
Molecular weight distribution of AAPs. Standards were insulin (5733 Da), bacitracin (1422 Da), Gly-Gly-Tyr-Arg (451 Da), and Gly-Gly-Gly (189 Da), respectively.

**Figure 3 nutrients-15-03731-f003:**
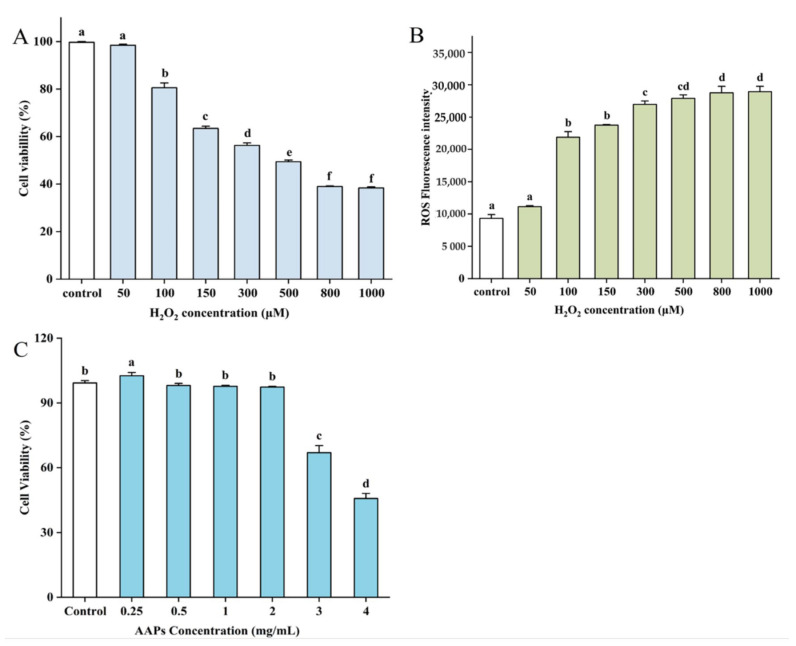
Effects of H_2_O_2_ on the viability of HepG2 cells (**A**), effects of H_2_O_2_ on ROS generation (**B**), and effects of AAPs on the viability of HepG2 cells (**C**). Results are shown as means ± standard deviation (SD, n = 3). Bar charts with the same superscript letters indicate that the difference is not significant (*p* > 0.05), while bar charts with different superscript letters indicate that the difference is significant (*p* < 0.05).

**Figure 4 nutrients-15-03731-f004:**
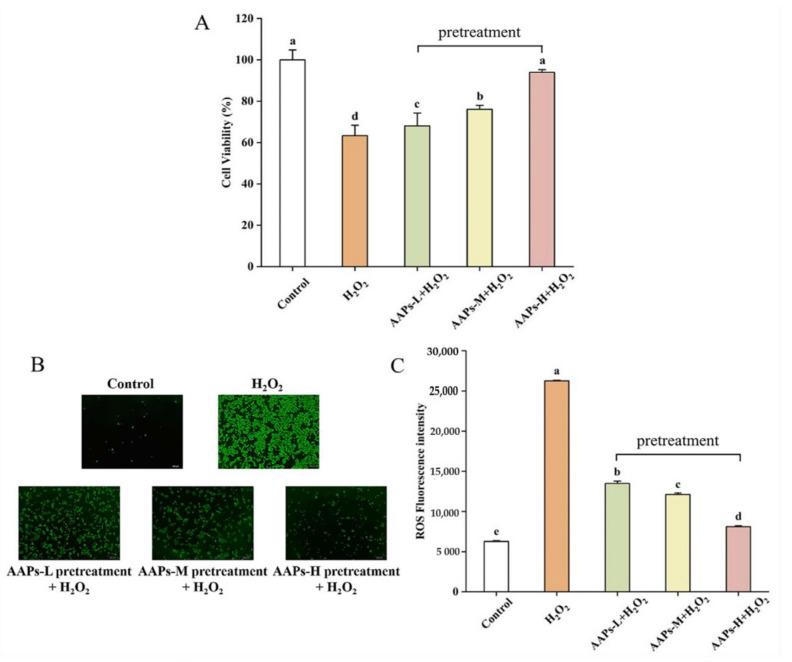
Effects of AAPs nutritional supplementation on the viability of HepG2 cells (**A**), fluorescence analysis of ROS (**B**), fluorescence intensity of different groups (**C**). Results were shown as means ± SD (n = 3). Bar charts with the same superscript letters indicate that the difference is not significant (*p* > 0.05), while bar charts with different superscript letters indicate that the difference is significant (*p* < 0.05).

**Figure 5 nutrients-15-03731-f005:**
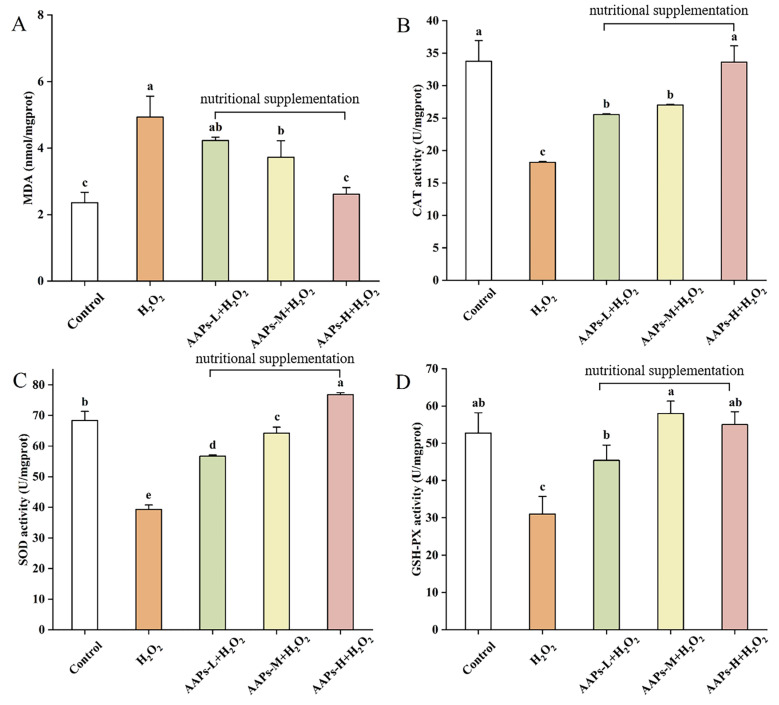
MDA (**A**), CAT (**B**), SOD (**C**), and GSH-Px (**D**) contents of different groups. Results were shown as means ± SD (n = 3). Bar charts with the same superscript letters indicate that the difference is not significant (*p* > 0.05), while bar charts with different superscript letters indicate that the difference is significant (*p* < 0.05).

**Figure 6 nutrients-15-03731-f006:**
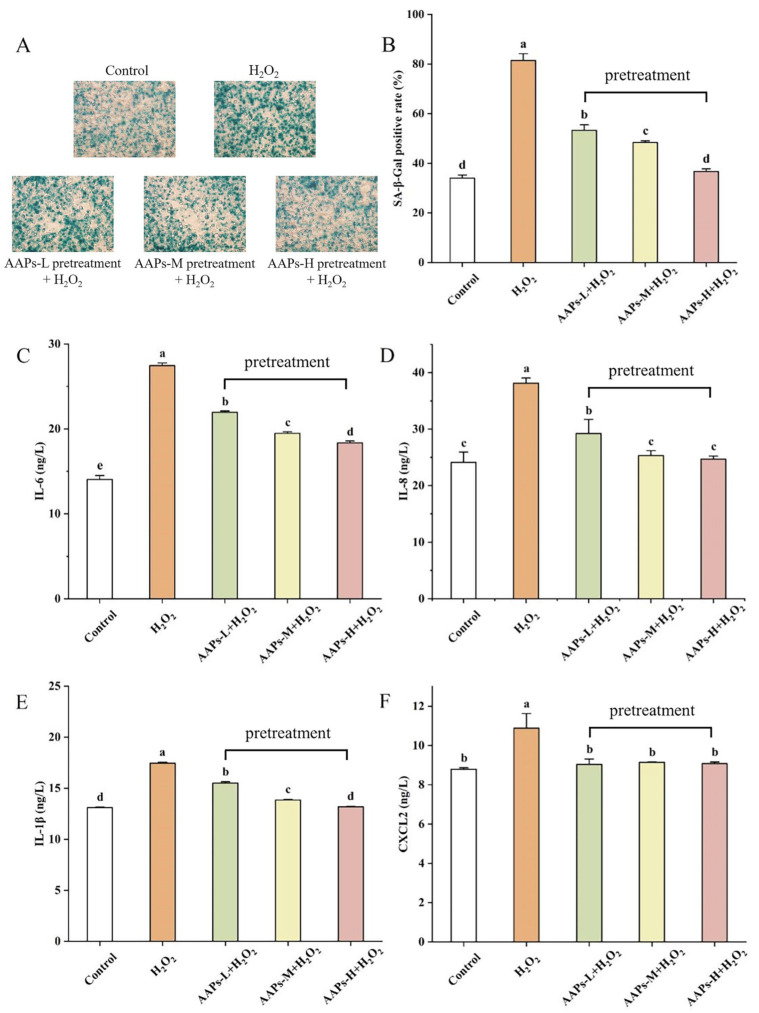
Effect of AAPs nutritional supplementation on SA-β-gal staining (**A**), positive straining rate of SA-β-gal (**B**), expression levels of typical senescence-associated secretory phenotypes (SASP), IL-6 (**C**), IL-8 (**D**), IL-1β (**E**), and chemokine CXCL2 (**F**). Results are shown as means ± SD (n = 3). Bar charts with the same superscript letters indicate that the difference is not significant (*p* > 0.05), while bar charts with different superscript letters indicate that the difference is significant (*p* < 0.05).

**Figure 7 nutrients-15-03731-f007:**
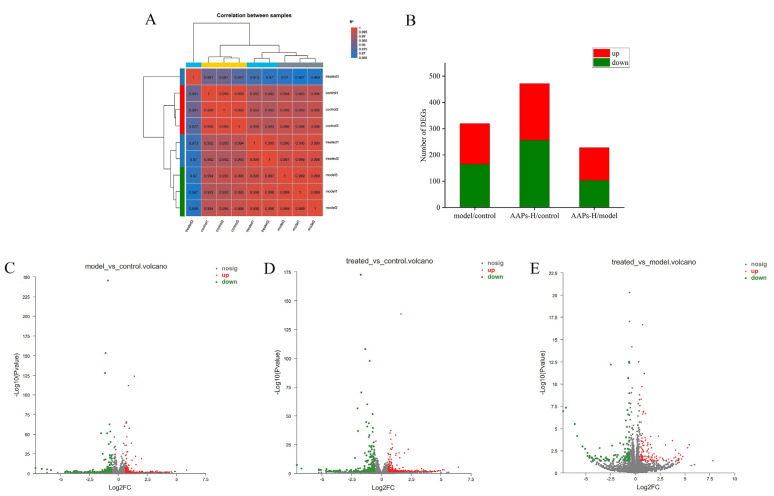
Correlation analysis of different groups (**A**), the numbers of DEGs between different groups (**B**), and the numbers of up−regulated and down−regulated genes in the model/control group (**C**), AAPs nutritional supplementation/control group (**D**), and AAPs nutritional supplementation/model group (**E**). Red, the gene expression is up−regulated; Green, the gene expression is down−regulated; Gray, the gene expression has not changed.

**Figure 8 nutrients-15-03731-f008:**
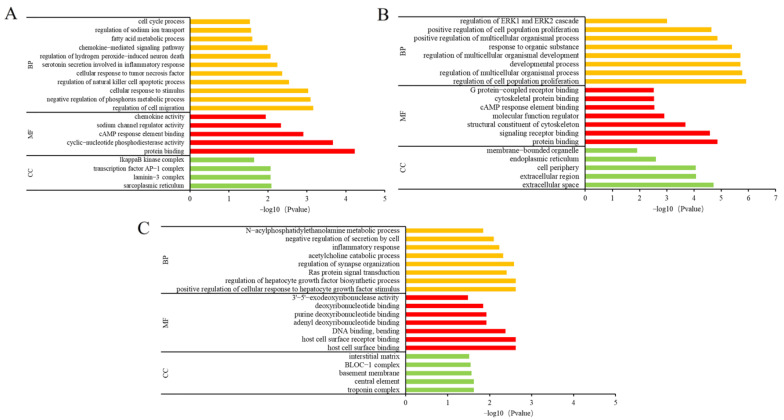
GO enrichment analysis of model/control group (**A**), AAPs nutritional supplementation/control group (**B**), and AAPs nutritional supplementation/model group (**C**).

**Figure 9 nutrients-15-03731-f009:**
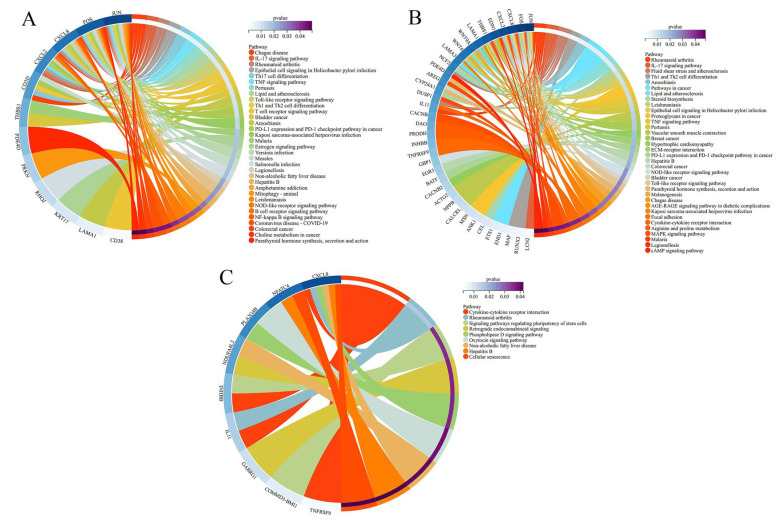
KEGG enrichment analysis of model/control group (**A**), AAPs nutritional supplementation/control group (**B**), and AAPs nutritional supplementation/model group (**C**).

**Figure 10 nutrients-15-03731-f010:**
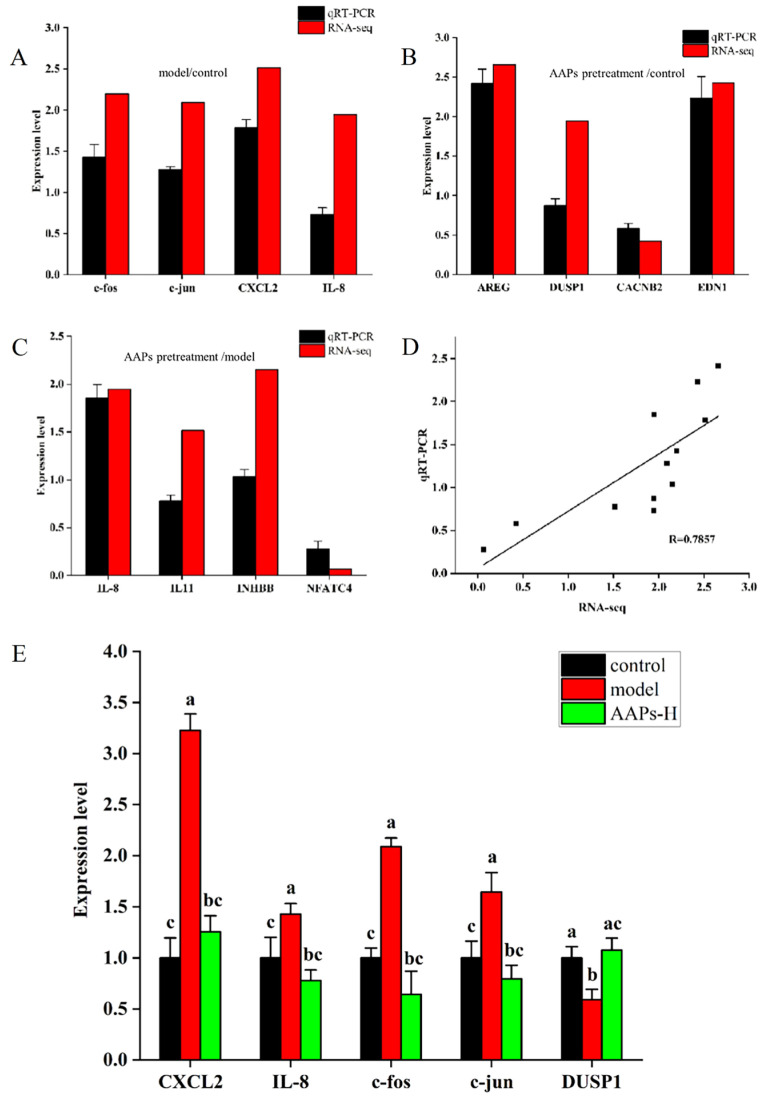
Comparison of qRT-PCR and RNA-seq in model/control group (**A**), AAPs nutritional supplementation/control group (**B**), and AAPs nutritional supplementation/model group (**C**), and correlation analysis (**D**). Comparison of expression levels of key genes in MAPK/NF-κB signaling pathway between different groups (**E**). R, correlation coefficient. Results are shown as means ± SD (n = 3). Bar charts with the same superscript letters indicate that the difference is not significant (*p* > 0.05), while bar charts with different superscript letters indicate that the difference is significant (*p* < 0.05).

**Figure 11 nutrients-15-03731-f011:**
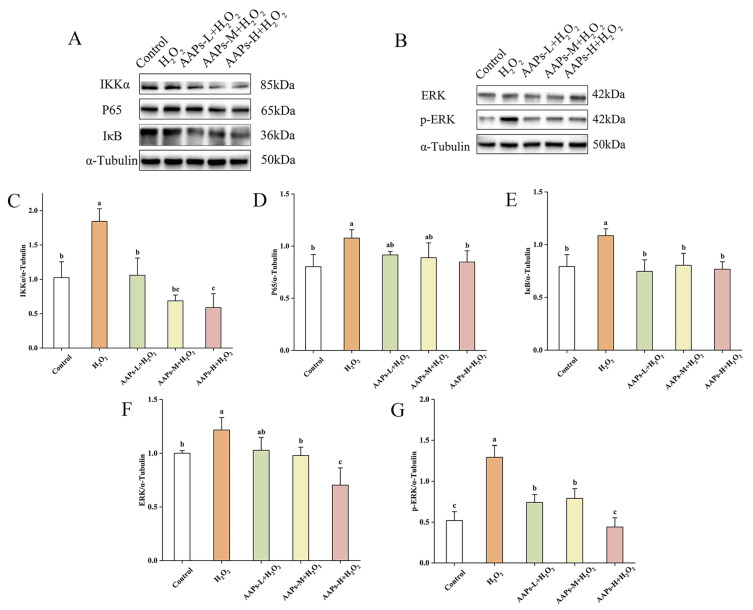
Western blot analysis of key proteins in NF-κB (**A**) and MAPK (**B**) signaling pathways, IKK (**C**), P65 (**D**), IκB (**E**), ERK (**F**), and *p*-ERK (**G**). Results are shown as means ± SD (n = 3). Bar charts with the same superscript letters indicate that the difference is not significant (*p* > 0.05), while bar charts with different superscript letters indicate that the difference is significant (*p* < 0.05).

**Table 1 nutrients-15-03731-t001:** Primer sequences of genes.

Gene	Forward Primer	Reward Primer
ACTB	5′-TGGAACGGTGAAGGTGACAG-3′	5′-AACAACGCATCTCATATTTGGAA-3′
c-fos	5′-GGGGCAAGGTGGAACAGTTAT-3′	5′-AGGTTGGCAATCTCGGTCTG-3′
c-jun	5′-GTGCCGAAAAAGGAAGCTGG-3′	5′-CTGCGTTAGCATGAGTTGGC-3′
CXCL2	5′-TTCACAGTGTGTGGTCAACAT-3′	5′-TCTCTGCTCTAACACAGAGGGA-3′
CXCL8	5′-ACTCCAAACCTTTCCACCCC-3′	5′-TTCTCAGCCCTCTTCAAAAACT-3′
AREG	5′-TACTCGGCTCAGGCCATTAT-3′	5′-TCCCGAGGACGGTTCACTAC-3′
CACNB2	5′-CCTGGGAGTAGAAGGTGGGA-3′	5′-GGAGTCTGCCGAACCATAGG-3′
DUSP1	5′-GCCATTGACTTCATAGACTCCA-3′	5′-ATGATGCTTCGCCTCTGCTT-3′
EDN1	5′-TTGAGATCTGAGGAACCCGC-3′	5′-GCTCAGCGCCTAAGACTGTT-3′
NFATC4	5′-GTCTTCCTTCCTCCTCCAGC-3′	5′-CTGAGTCCAGTTCTTCCCCC-3′
IL11	5′-GGACATGAACTGTGTTTGCCG-3′	5′-GAGGGTCTGGGGAAACTCG-3′
INHBB	5′-GCGAGAACCCTCAACTGACA-3′	5′-ACCGCATCCATTTGCTGGTA-3′

**Table 2 nutrients-15-03731-t002:** Amino acid compositions (%) of APPs.

Item	AAPs (%)
Asparticacid (Asp)	9.28 ± 1.23
Threonine (Thr)	3.00 ± 0.06
Serine (Ser)	4.35 ± 0.09
Glutamicacid (Glu)	19.17 ± 0.19
Glycine (Gly)	10.57 ± 0.46
Alanine (Ala)	4.69 ± 0.77
Cysteine (Cys)	7.81 ± 0.21
Valine (Val)	3.87 ± 0.03
Methionine (Met)	1.39 ± 0.50
Isoleucine (Ile)	3.76 ± 0.09
Leucine (Leu)	6.39 ± 0.10
Tyrosine (Tyr)	3.14 ± 0.36
Phenylalanine (Phe)	4.56 ± 0.27
Histidine (His)	3.26 ± 0.07
Lysine (Lys)	6.31 ± 0.04
Arginine (Arg)	8.43 ± 0.21

Values are expressed as means ± SD (n = 3).

**Table 3 nutrients-15-03731-t003:** An overview of the RNA-Seq statistics.

Sample	Raw Reads	Raw Bases	Clean Reads	Clean Bases	Error Rate (%)	Q20 (%)	Q30 (%)
control 1	60,098,740	9,074,909,740	58,473,192	8,587,028,318	0.0253	97.89	93.99
control 2	54,787,316	8,272,884,716	53,551,530	7,888,155,174	0.0246	98.18	94.63
control 3	62,491,860	9,436,270,860	60,960,432	8,948,341,313	0.0252	97.92	94.01
model 1	55,742,576	8,417,128,976	54,448,302	7,991,536,234	0.0248	98.11	94.43
model 2	70,035,438	10,575,351,138	67,992,638	9,996,752,010	0.0250	98.01	94.29
model 3	59,654,220	9,007,787,220	57,961,824	8,550,212,323	0.0249	98.02	94.29
treated 1	51,838,286	7,827,581,186	50,408,474	7,427,178,279	0.0248	98.06	94.39
treated 2	59,391,338	8,968,092,038	57,718,770	8,513,014,983	0.0251	97.97	94.17
treated 3	53,077,068	8,014,637,268	51,623,830	7,544,501,778	0.0248	98.10	94.50

## Data Availability

The data presented in this study are available on request from the corresponding author.

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
