# Peer review of "Auricularia auricula Peptides Nutritional Supplementation Delays H2O2-Induced Senescence of HepG2 Cells by Modulation of MAPK/NF-κB Signaling Pathways"

_nutrients, 2023, doi:10.3390/nu15173731_

Round 1

Reviewer 1 Report

General comments:

This paper describes the effect of peptides extracted from Auricularia auricula  on delaying H2O2-induced senescence of HepG2 cells by modulation of 3 MAPK/NF-κB signaling pathways. The study is well carried out, mostly well written and is largely free of errors.

Specific comments:

1. In Table 2, the number of replicates from which the SD is calculated should be given.

2. In Figure 2, the numerical values of the standards are given on the chromatogram but not identified to the standards given under the figure label.

3. The peaks in Figure 2 are poorly resolved. The authors should state the uncertainty this introduces to their quantitative data.

4. In L 146 does “three samples” refer to “three replicate samples”?

5. In Figure 3, the number of replicates should be given.

Quality of English is good.

Reviewer 2 Report

1. How do bioactive peptides from edible fungi differ from those derived from animals in terms of absorption and availability in the body?

2. What is the potential role of Auricularia auricula peptides (AAPs) in reducing MDA levels and increasing SOD levels in HepG2 cells, and how does this relate to anti-aging effects?

3. Could you elaborate on the specific mechanisms by which oxidative stress and inflammation contribute to the aging process, as well as how they are interconnected?

4. What are the proposed twelve hallmarks of aging, and how does inflamm-aging relate to the development of age-related disorders?

5. Can you explain how ROS (reactive oxygen species) function as signaling factors at lower levels and their impact on gene expression?

6. What is the significance of MDA as an end product of lipid peroxidation, and how does it contribute to oxidative damage at the cellular level?

7. How do antioxidant enzymes such as CAT, SOD, and GSH-Px maintain the balance between oxidants and antioxidants in organisms?

8. What is the relationship between SA-β-gal activity and cell senescence, and why is detection of β-galactosidase activity at pH 6.0 important?

9. Could you explain the concept of SASP (senescence-associated secretory phenotype) and how senescent cells contribute to inflammation and senescence in neighboring cells?

10. How does the NF-κB signaling pathway contribute to the regulation of SASP factors and the overall inflammatory response in senescent cells?

11. How do extracellular mediums and signaling cascades, such as MAPK (mitogen-activated protein kinase), regulate NF-κB activation and inflammatory factors?

12. What are the unique characteristics of the HepG2 cell line that make it suitable for anti-aging studies, and why is it commonly used in biochemical and nutritional research?

13. In the context of oxidative damage-induced cell senescence, how does the H2O2-induced HepG2 model provide insights into potential protective effects of Auricularia auricula peptides on the liver?

14. What specific methods were used for the isolation and preparation of Auricularia auricula peptides (AAPs), and how was their protein content, amino acid composition, and molecular weight determined?

15. How does the use of alkaline solubilization and acid deposition methods impact the isolation process of Auricularia auricula protein and subsequent peptide production?

16. What is the significance of the 60% degree of hydrolysis achieved in the preparation of AAPs, and how was the final product collected and characterized?

17. Could you elaborate on the rationale behind using HepG2 cells as a model for studying the effects of AAPs on cell viability and oxidative stress?

18. How was the H2O2-induced HepG2 senescent model established, and what concentrations of H2O2 were tested for evaluating its effects on cell viability and ROS levels?

19. Can you explain the principle behind the CCK-8 assay used to measure cell viability, and how were the effects of AAPs and H2O2 on HepG2 cell viability quantified?

20. What is the significance of the DCFH-DA assay in measuring ROS levels, and how does it relate to evaluating the protective effect of AAPs nutritional supplementation on cell senescence?

21. How were the levels of MDA (malondialdehyde), SOD (superoxide dismutase), CAT (catalase), and GSH-Px (glutathione peroxidase) determined in the cells, and why are these antioxidant enzymes important?

22. How was SA-β-gal (senescence-associated β-galactosidase) staining performed, and why is it a common method for identifying senescent cells?

23. Can you explain the concept of the senescence-associated secretory phenotype (SASP) and how the concentrations of IL-1β, IL-6, IL-8, and CXCL2 were measured to evaluate the effects of AAPs and H2O2 on SASP?

24. What is the purpose of library construction and sequencing in this study, and how was the RNA-seq data analyzed to gain insights into the effects of AAPs and H2O2 on gene expression?

25. How was the quantitative real-time PCR (qRT-PCR) method used to confirm the transcriptome data and validate the expression levels of selected genes?

26. What was the rationale behind using the ACTB gene as an internal control in the qRT-PCR experiments, and how were the expression levels of target genes determined using the 2−ΔΔCt method?

27. How was the protein content of AAPs determined, and what is the significance of its 62.62% value in terms of the potential antioxidant activity?

28. Can you elaborate on the relationship between the amino acid composition of AAPs and their antioxidant capacity, particularly with respect to aromatic, acidic, and basic amino acids?

29. What is the rationale behind the selection of specific molecular weight ranges (less than 1000 Da and 1000-3000 Da) for AAPs, and how does this distribution contribute to their antioxidant activity?

30. How was the optimal concentration of H2O2 (150 μM) determined for inducing oxidative stress in HepG2 cells, and what were the observed effects on cell viability and ROS levels?

31. Could you explain the reasoning behind selecting concentrations of 0.25, 0.5, 1, and 2 mg/mL for AAPs, and why were the 0.5, 1, and 2 mg/mL concentrations chosen for further research?

Good luck!

required minor changes. 

Reviewer 3 Report

The authors submitted an interesting original research manuscript which deals with anti-aging effects of edible mushroom Auricularia auricular on HepG2 cell cultures. Since so called civilization diseases associate with aging population of various countries become gradually a serious problem for health care systems, the topic of the manuscript is timely and important.

The authors isolated a fraction of Auricularia auricula peptides (AAPs) by enzymatic hydrolysis of the mycelium proteins. After the proten fraction characterization, the proteins were investigated on HepG2 cells and explored the activities against the cell again induced by hydrogen peroxide.  The authors used well-established and adequate methods usually employed in this type of biochemical or molecular biological research.

The experimental design and planning were nearly adequate and the results are supported by the experimental data. Furthermore, the results are presented in many illustrative figures and plots. Nevertheless, I see a weakness that the authors did not compare the AAps effects with a standard antioxidant with well known anti-aging effects such as rutin, curcumin or other standard ROS scavenger. If the mentioned comparison was not performed as a apparel experiment, I recommend discussing similar experiments in discussion.
